# miRNA Clusters with Up-Regulated Expression in Colorectal Cancer

**DOI:** 10.3390/cancers13122979

**Published:** 2021-06-14

**Authors:** Paulína Pidíková, Iveta Herichová

**Affiliations:** Department of Animal Physiology and Ethology, Faculty of Natural Sciences, Comenius University in Bratislava, Ilkovičova 6, 842 15 Bratislava, Slovakia; paulina.pidikova@uniba.sk

**Keywords:** metastases, survival, oncogenes, tumour suppressors, miR-17/92a-1, miR-183/96/182

## Abstract

**Simple Summary:**

As miRNAs show the capacity to be used as CRC biomarkers, we analysed experimentally validated data about frequently up-regulated miRNA clusters in CRC tissue. We identified 15 clusters that showed increased expression in CRC: miR-106a/363, miR-106b/93/25, miR-17/92a-1, miR-181a-1/181b-1, miR-181a-2/181b-2, miR-181c/181d, miR-183/96/182, miR-191/425, miR-200c/141, miR-203a/203b, miR-222/221, mir-23a/27a/24-2, mir-29b-1/29a, mir-301b/130b and mir-452/224. Cluster positions in the genome are intronic or intergenic. Most clusters are regulated by several transcription factors, and by long non-coding RNAs. In some cases, co-expression of miRNA with other cluster members or host gene has been proven. miRNA expression patterns in cancer tissue, blood and faeces were compared. The members of the selected clusters target 181 genes. Their functions and corresponding pathways were revealed with the use of Panther analysis. Clusters miR-17/92a-1, miR-106a/363, miR-106b/93/25 and miR-183/96/182 showed the strongest association with metastasis occurrence and poor patient survival, implicating them as the most promising targets of translational research.

**Abstract:**

Colorectal cancer (CRC) is one of the most common malignancies in Europe and North America. Early diagnosis is a key feature of efficient CRC treatment. As miRNAs can be used as CRC biomarkers, the aim of the present study was to analyse experimentally validated data on frequently up-regulated miRNA clusters in CRC tissue and investigate their members with respect to clinicopathological characteristics of patients. Based on available data, 15 up-regulated clusters, miR-106a/363, miR-106b/93/25, miR-17/92a-1, miR-181a-1/181b-1, miR-181a-2/181b-2, miR-181c/181d, miR-183/96/182, miR-191/425, miR-200c/141, miR-203a/203b, miR-222/221, mir-23a/27a/24-2, mir-29b-1/29a, mir-301b/130b and mir-452/224, were selected. The positions of such clusters in the genome can be intronic or intergenic. Most clusters are regulated by several transcription factors, and miRNAs are also sponged by specific long non-coding RNAs. In some cases, co-expression of miRNA with other cluster members or host gene has been proven. miRNA expression patterns in cancer tissue, blood and faeces were compared. Based on experimental evidence, 181 target genes of selected clusters were identified. Panther analysis was used to reveal the functions of the target genes and their corresponding pathways. Clusters miR-17/92a-1, miR-106a/363, miR-106b/93/25 and miR-183/96/182 showed the strongest association with metastasis occurrence and poor patient survival, implicating them as the most promising targets of translational research.

## 1. Introduction

The present systematic review is focused on clusters of micro RNAs (miRNAs) with up-regulated expression in colorectal cancer (CRC) tissue showing potential for use as biomarkers. The improvement of CRC diagnosis and treatment is still a pressing issue, as CRC is the third most commonly diagnosed cancer and the second most frequent cause of cancer death worldwide. The highest CRC incidence rate has been revealed in Europe (predominantly Western and Northern Europe), Australia, New Zealand, North America and East Asia. The incidence of CRC has increased mainly in the younger population (up to 55 years) in last two decades [1].

miRNAs are a class of small non-coding RNAs with a length of approximately 20 bp [2,3]. Several miRNAs are already used in clinics as a diagnostic tool and/or in cancer treatment [4]. Biosynthesis of miRNAs consists of two processing steps. First, the primary miRNA transcript (pri-miRNA) is processed in the nucleus by the protein complex microprocessor containing the enzyme Drosha and the cofactor DGCR8 to generate a precursor miRNA (pre-miRNA). The second processing step, executed by the enzyme Dicer and its cofactors TAR (HIV-1) RNA binding protein 2 (TARBP2) and protein activator of interferon induced protein kinase EIF2AK2 (PRKRA), occurs after pre-miRNA transport to the cytoplasm. Expression of Drosha, Dicer and DGCR8 has been shown to be up-regulated in colorectal adenocarcinoma, compared with non-neoplastic tissue [5]. Moreover, increased expression of the enzymes Dicer [6], Drosha and cofactor TARBP2 [7] was associated with worse survival. The role of Dicer in CRC progression is also supported by the finding that oxaliplatin treatment is associated with a decrease in Dicer levels in human CRC cells [8]. However, enzymes of the miRNA biosynthesis pathway do not seem to be the most important factor determining the up-regulation of miRNA expression in cancer tissue [6], and transcription regulation seems to be more important in this respect. The rate of expression depends on regulatory regions of miRNA genes located in intergenic areas or host genes of intragenic miRNAs. Intragenic miRNAs are located in intron or exon areas of their host genes, and thus, they make coordinated transcription of host gene and miRNA possible [9,10,11].

Therefore, we performed a screen of miRNA clusters up-regulated in CRC tissue and focused on the correlation of their expression with clinicopathological characteristics, patient prognosis, response to therapy and survival. miRNA locations in the genome were analysed to assess possible modes of regulation of their expression. The functions of the target genes of selected up-regulated miRNA clusters in human CRC cell lines were also studied using in silico research tools to determine if their up-regulation contributes to CRC progression. On the basis of the obtained results, modes of regulation and functions of specific miRNA clusters were determined.

## 2. Materials and Methods

### 2.1. Literature Search Strategy

The PubMed and Scopus databases were searched to find research papers referring to miRNA expression screening in colorectal cancer tissue. Research articles published up to 12 September 2020 were collected. The search terms used were (“colorectal cancer”) AND ([miRNA expression profile] OR [microRNA expression profile]) AND ([upregulation] OR [up-regulation]). In preliminary screening, duplicate articles were omitted.

#### 2.1.1. Eligibility Criteria for Articles

The criteria for inclusion of articles were as follows:Studies published in the English languageStudies with samples of human tumour tissue from patients with CRCStudies where the miRNA expression profile of the CRC tumour tissue was compared with that of normal tissueStudies with available expression data about up-regulated miRNAs

The criteria for exclusion of articles were as follows:Reviews, book chapters, meta-analysis, and systematic reviewsStudies using only animal models and cell linesStudies where only plasma or serum samples from patients with CRC were analysed.

#### 2.1.2. Eligibility Criteria for Selecting miRNAs and miRNA Clusters

For further analyses, only those miRNA clusters where the majority of their members had been reported in three or more studies were selected. In the next step, the genome location of selected miRNA clusters was investigated. Only miRNA clusters located on the same DNA strand, where the distance between miRNAs in one cluster was lower than 10 kbp, were included in the study.

### 2.2. Searching for Further Data about miRNAs

In the next step, we searched the PubMed database for articles about selected miRNAs involved in clusters that met the inclusion criteria. We focused on studies analysing the expression of selected miRNAs in human CRC tissues, the circulation or stool. Associations of miRNA expression with clinicopathological characteristics and target genes of selected miRNAs, validated in human CRC cells lines, were also extracted from these articles. In this way, 418 references were included into study.

### 2.3. Classification of Target Genes by Panther Analysis

In vitro experimentally validated target genes of miRNA clusters up-regulated in CRC were classified with the Gene Ontology (GO) Analysis Panther Classification System (GO database Version 16.0 released 1 December 2020) [12]. Genes were grouped according to four classification categories: molecular function, biological process, protein class and pathway. An overrepresentation test was also used to identify enriched categories in each classification. Statistical significance was evaluated using Fisher’s exact test and the calculated false discovery rate (FDR). The GO-Slim and Panther Pathway categories with *p* < 0.05 and an FDR < 0.05 were considered statistically significant.

## 3. Results

### 3.1. Characteristic of Studies Involved in Meta-Analysis

In total, 225 studies were found using the PubMed database, and 174 studies were found using Scopus. In preliminary screening, we omitted 99 duplicated articles. In the next step, according to the inclusion and exclusion criteria, we included 47 articles in our study. The searching process is depicted in Figure 1. The included studies are listed in Appendix A. A total of 496 up-regulated miRNAs were reported in the selected studies; among them, 107 mature miRNAs were reported in three or more studies.

From these 107 miRNAs, we selected 64 mature miRNAs which were located in clusters. According to the selection criteria, 15 miRNA clusters with up-regulated miRNA in CRC tissues were included in the study.

The miRNA clusters included in the study are listed below. Cluster miR-106a/18b/20b/19b-2/92a-2/363 is abbreviated in further text as miR-106a/363 and cluster miR-17/18a/19a/20a/19b-1/92a-1 is abbreviated as miR-17/92a-1.

miR-106a/18b/20b/19b-2/92a-2/363 (miR-106a/363)miR-106b/93/25miR-17/18a/19a/20a/19b-1/92a-1 (miR-17/92a-1)miR-181a-1/181b-1miR-181a-2/181b-2miR-181c/181dmiR-183/96/182miR-191/425miR-200c/141miR-203a/203bmiR-222/221miR-23a/27a/24-2miR-29b-1/29amiR-301b/130bmiR-452/224

Next, we closely analysed the selected miRNAs according to their location in the human genome, their expression in human CRC samples and their association with clinicopathological characteristics. Only those target genes of miRNAs that have been validated in experimental studies were included in the study. Afterwards, target genes were classified according to GO-Slim Panther analyses, and an overrepresentation test [12] was used to reveal significantly enriched categories in each Panther classification category.

### 3.2. Genomic Location of Selected Clusters and Regulation of Their Expression

Comprehensive analysis of clusters with up-regulated expression in CRC tissue revealed that eight clusters are located in intronic areas of protein-coding genes. Another seven clusters are located in intergenic regions. This distribution agrees with previously published studies, which revealed that approximately one half of miRNAs in the human genome are located in intronic areas of genes [11,13]. On the other hand, our previous study revealed that miRNA clusters with down-regulated expression are predominantly located in intronic areas of genes [14]. All selected clusters and their host genes are listed in Table 1.

Three clusters, miR-106a/363, miR-222/221 and miR-452/224, are located on the X chromosome. Chromosome 7 hosts three clusters, miR-106b/93/25, miR-183/96/182 and miR-29b-1/29a, and two clusters, miR-23a/27a/24-2 and miR-181c/181d, are located on chromosome 19.

An earlier study showed that transcription of intronic miRNAs is regulated by the promoter of their host gene, and intronic miRNA expression is thus frequently correlated with that of the host gene [9]. However, several later studies revealed that co-expression of intronic miRNAs with their host genes may vary according to evolutionary miRNA conservation or localization on the DNA strand. Evolutionarily conserved miRNAs were more frequently co-expressed with their host genes, compared with evolutionarily non-conserved miRNAs [16,21]. Additionally, antisense miRNAs localised on the opposite strand to their host genes were less frequently co-expressed with their host genes, compared with sense miRNAs [21]. Intragenic miRNAs are predominantly independent transcription units, and they have their own transcription regulatory elements [25]. In Table 1, selected miRNA clusters are listed along with a characterization of the transcription start sites (TSS) of miRNA genes. Transcription of intronic miRNA genes may be regulated by their host gene’s TSS (characterised as TSS-dependent), or intronic miRNA genes may have their own TSS (independent TSS). Co-expression with the host gene was analysed in human CRC tissues (Table 1).

The most miRNA clusters in the human genome contain two to six miRNA members [32]. All clusters included in our study fit within this range. Ten of the selected clusters contain two miRNA members each, three of the selected clusters contain three miRNA members each, and two clusters have six miRNA members.

Transcription factors involved in miRNAs regulation usually also influence the expression of protein-coding genes. However, intergenic and intronic miRNAs seem to be preferentially regulated by different sets of transcription factors [10]. In our study, attention is given to three transcription factors supported by strong experimental evidence, zinc finger E-box binding homeobox 1 (ZEB1), MYC proto-oncogene, bHLH transcription factor (MYC) and p53, which have been validated as regulators of several selected miRNA clusters in human CRC cell lines [33,34]. ZEB1, MYC and p53 mediated regulation of miRNA transcription is described in detail in the discussion.

Epigenetic DNA modifications, such as methylation, are important inhibitory factors of gene expression, and several methylation markers have been proposed as prognostic epigenetic markers for CRC [35]. Epigenetic changes are also involved in the regulation of miRNA expression. In particular, promoter methylation with an inhibitory influence on the expression of clusters miR-200c/141 [36,37] and miR-363 [38] has been proven.

### 3.3. Regulation of miRNA Expression by Long Non-Coding RNAs

RNA transcripts, not translated into proteins, with sequences longer than 200 nucleotides are commonly classified as long non-coding RNAs (lncRNA), can be divided into several subclasses according to their functions and origins. The most studied groups of lncRNAs are long intergenic ncRNAs (lincRNAs), antisense RNAs (asRNAs), pseudogenes, circular RNAs (circRNAs) and intronic lncRNAs [39].

Mature lncRNAs are folded into complex secondary and tertiary structures with binding domains for interaction with proteins, DNA and other RNA molecules [40]. Interplay between lncRNAs and miRNAs has been observed at several regulatory levels [9,39]. Firstly, lncRNAs may serve as host genes of miRNAs [9]. Analysis of 232 miRNAs revealed that 10% of analysed miRNAs are located in introns of lncRNAs. Thirty miRNA genes which overlap with exons of non-coding RNAs were also identified [22]. Biogenesis of miRNAs transcribed from lncRNA host genes differs from canonical miRNA biogenesis in several aspects, such as transcription termination or polyadenylation of miRNA gene transcript. In our study, six host genes of selected clusters are also classified as lncRNAs—MIR17HG, MIR181AHG, MIR200CHG, MIR222HG, MIR23AHG and LINC-PINT (Table 1). Another function of lncRNAs is binding to miRNAs with complementary sequences and repressing miRNA–mRNA interaction (competing endogenous RNAs). The regulatory function of lncRNAs contributes significantly to the cancer development and progression and numerous lncRNAs with tumour-promoting or tumour-suppressing functions in colorectal cancer have already been identified [40].

Our analysis revealed 34 lncRNAs validated as sponges for miRNAs from selected clusters in CRC cell lines (Table 2). The most of the detected lncRNAs show an inhibitory influence only with respect to one miRNA in colorectal cancer cells, but our analysis also revealed three lncRNAs that sponge multiple miRNAs or even a whole miRNA cluster. The lncRNA CCAT1 sponges miRNAs from two homolog clusters, miR-181a-1/181b-1 and miR-181a-2/181b-2 [41,42]. The lncRNA ATB sponges the expression of both miRNAs from cluster miR-200c/141 [43,44].

Nearly one third of detected lncRNAs are classified as asRNAs (Table 2). Antisense RNAs are located on the opposite strand of a protein-coding DNA or protein non-coding genes and are involved in the modulation of miRNA-mediated inhibition of expression of a gene in their close proximity [45]; e.g., ZEB1 antisense RNA 1 (asRNA ZEB1-AS1) influences the expression of the transcription factor ZEB1 by sponging miR-141-3p, which inhibits ZEB1 expression [46,47].

Intronic lncRNAs may also be involved in the regulation of the expression of genes located in their proximity via interactions with miRNA, e.g., lncRNA AGER-1, located in the intron of the AGER-coding gene (advanced glycosylation end-product specific receptor), inhibits interaction of miR-182 with AGER mRNA by miRNA sponging [48]. Newly identified lncRNA generated as a splicing variant of gene LINC-PINT was shown to sponge a member of the miR-29b-1/29a cluster, miR-29b, located in the same host gene in CRC cells [29].

### 3.4. miRNA Families

A miRNA family includes homolog miRNAs with identical seed sequences. A seed sequence is a region located in the 5′ end of mature miRNA with a length of seven to eight nucleotides. Binding of miRNA to its target mRNA relies predominantly on the miRNA seed sequence, but the 3′ and central regions of the miRNA sequence are also of importance in mRNA recognition [84]. Thus, the inhibitory potential of miRNAs from the same family with identical seed sequences may differ depending on RNA context in the proximity of seed sequence [85].

miRNA clusters may contain miRNAs from different families and, *vice versa*, miRNAs from one family can be located in different clusters. More than half of all human miRNA clusters contain homolog miRNAs (from one family) [86]. This distribution was also observed among the 15 clusters analysed in this study, where seven clusters contained only miRNAs classified in one family (Table 3). Evolutionary analysis showed that miRNA clusters with homolog miRNA seed sequences were mainly created by gene duplication [32]. On the other hand, five clusters contained miRNAs only from distinct families.

The up-regulated miRNA clusters selected for our study contain miRNAs from 21 miRNA families. A substantial number of the selected miRNAs belong to the family miR-17-5p/20-5p/93-5p/106-5p/519-3p, which contains six miRNAs located in three homolog clusters: miR-106a/363, miR-106b/93/25 and miR-17/92a-1 (Table 3).

Nearly 20% of miRNA clusters in the human genome contain mature miRNAs that can be generated from more than one pre-miRNA [86]. The miRNA clusters analysed in our study contain 41 pre-miRNAs, which are processed into 37 mature miRNAs. Four mature miRNAs are processed from two precursors each, which are located in several clusters. Mature miR-181a and miR-181b are processed from their precursor miRNAs included in two clusters: miR-181a-1/181b-1 and miR-181a-2/181b-2. Similarly, mature miR-92a and miR-19b are processed from precursors included in two miRNA clusters: miR-17/92a-1 and miR-106a/363. Mature miR-24 is also processed from two pre-miRNAs from two different clusters, but we selected for this study only cluster miR-23a/27a/24-2. Its homologue cluster miR-23b/27b/24-1 did not fit our criteria for miRNA cluster selection. Similarly, only one precursor for miR-29b from the miR-29b-1/29a cluster was included in the analyses, and the miR-29b-2/29c cluster was omitted.

### 3.5. Expression of miRNA Clusters in Tissues, the Circulation and Stool of Patients with CRC

Up-regulation of selected miRNA expression in tissues and the circulation of CRC patients was verified by a set of studies obtained by a search in the PubMed database. The expression of 14 miRNA clusters was predominantly increased in CRC tumour tissues, compared with adjacent tissues. Nearly all members of the miR-106a/363 cluster were identified in the majority of the studies as being up-regulated in CRC tissue (Appendix A).

Tumour biopsy is an invasive procedure, and in many instances, it is difficult to obtain a sufficient amount of tissue for genome profiling and gene expression analysis. Liquid biopsy is a new diagnostic concept based on analyses of cell-free DNA, RNA and other molecules in the circulation [88]. miRNAs in the circulation are encapsulated in exosomes and microvesicles. Extracellular vesicles secreted by cells could be classified according to their size and biogenesis into six categories: exosomes, microvesicles, ectosomes, large oncosomes, exosome-like vesicles and apoptotic vesicles [89]. Extracellular vesicles containing various classes of bioactive molecules—DNA, RNA, proteins and lipids—seem to be an important tool for homotypic and heterotypic communication of cells in tumours [90]. However, there is also a significant amount of free circulating miRNA, which is associated with the Argonaut2 protein [91].

The release and uptake of exosomes is regulated by complex mechanisms, which may be modified by the tumour microenvironment; e.g., uptake of exosomes is regulated by pH as lower pH induces an increase in the uptake and release of exosomes. However, there are indications that at least some of the miRNAs in the circulation originate in white blood cells, and thus, the levels of miRNAs in the plasma may vary according to changes in the leukocyte count [92].

Tumour cells actively reprogram stromal cells and cells of the immune system through exosome secretion [93,94]. Genes and miRNAs in exosomes may modify the growth or invasiveness of recipient cells [93]. The influence of microvesicles derived from CRC cell lines on monocytes seems to depend on the degree of monocyte differentiation [95].

Most of the selected miRNA clusters showed up-regulated expression in the plasma and serum, compared with the control group. The expression of all miRNAs from the clusters miR-17/92a-1, miR-181a-1/181b-1, miR-181a-2/181b-2, miR-181c/181d, miR-183/96/182, miR-191/425, miR-200c/141, miR-203a/203b, miR-222/221 and miR-301b/130b was predominantly increased in plasma, serum or isolated exosomes of patients with CRC, compared with the control group. Information about the up- or down-regulation of the analysed miRNA clusters in the circulation is given in Appendix A.

Surgical removal of the tumour led to decreased plasma expression of miR-106a-5p [96], miR-18a, miR-17-3p and miR-92a [97] from two homolog clusters: miR-17/92a-1 and miR-106a/363. Expression of miR-23a and miR-20a in plasma was lower after radical surgical removal of the tumour [98] and expression of miR-29a and miR-92a decreased in post-operative plasma [99]. Plasma miR-182 levels decreased one month after radical liver metastasectomy, compared with that in preoperative plasma [100]. Expression of miR-29b-3p was decreased in postoperative plasma, compared with that in preoperative plasma, and after dividing patients by gender, this difference remained significant in both groups [101]. After surgical removal of the tumour, the expression of miR-23a was decreased in serum exosomes [102] and plasma extracellular vesicles [103] compared with preoperative levels. Similarly, tumour resection led to decreased expression of miR-106a and miR-17-3p in post-operative serum samples, compared with preoperative serum [104]. Expression of miR-182 was decreased in postoperative serum, compared with serum collected before operation. However, in patients with postoperative CRC recurrence, re-elevation of miR-182 serum levels was observed [105].

The abovementioned evidence implicates an association between miRNA levels in the circulation and pre- and post-surgery levels of miRNA in the colorectum. Surgical removal of the tumour and/or metastases frequently results in decreases in associated miRNAs in the circulation. Changes in miRNA levels in the plasma were more prominent after radical surgical removal of the tumour, compared with palliative tumour removal (liver metastases were not removed) [98].

Current widely used methods for early diagnosis of CRC include the faecal occult blood test (FOBT) and the faecal immunochemical test [106]. Isolation of DNA from colonocytes in faeces enabled DNA analyses, such as analyses of gene mutations and epigenetic changes related to CRC development. Analyses of the expression of genes and other non-coding RNAs from stool is also a potential tool for non-invasive CRC diagnosis [107,108,109]. Widely used non-invasive screening methods include the faecal blood test, detecting the presence of haemoglobin in stool, the faecal immunochemical test and a test based on DNA isolated from colonocytes in faeces [106,107,110].

miRNAs are also detectable in stool samples, and screening of faecal miRNA expression may be a useful tool for the diagnosis of intestinal diseases. The stability of faecal miRNAs depends on their origin, as miRNAs in colonocytes and exosomes are more resistant to degradation by RNase than free miRNAs [111]. Analysis of human CRC tumours and adjacent normal mucosa specimens showed that colonocytes in the mucocellular layer over CRC tumours were more abundant than those in the mucocellular layer over the normal mucosa [112]. The levels of all members of the miR-17-92a-1 cluster were increased in the stools of CRC patients, compared with controls. On the other hand, the levels of members of the miR-29b/miR-29a cluster were decreased (Table 4).

miRNAs from the homolog clusters miR-17/92a-1 and miR-106a/363 were proposed as potential biomarkers in stool samples. The levels of miR-18a-5p, miR-19a-3p, miR-19b-3p, miR-20a-5p, miR-92a-3p and miR-106a were decreased in stool samples collected 12–30 months after surgery, compared with preoperative samples. Similarly, the levels of miR-20a-5p and miR-141 in postoperative samples were normalised to those in healthy control samples. On the other hand, the levels of miR-92a-3p remained higher compared to control [114].

Other biological substances that can be collected using non-invasive methods and may be used as a source of several types of biomarkers for CRC diagnosis include saliva, urine and colonic mucus [109]. A study from 2019 revealed that the expression of miR-29a-3p was increased in the saliva of patients with CRC compared with controls. The levels of miR-29a-3p in saliva also showed sex-dependent differences, as expression in the saliva of men was increased, compared with that in women [120].

### 3.6. Association of miRNA Clusters Expression with Clinicopathological Characteristics

Detection of CRC in its early stages is associated with a better prognosis for patients. The initial stages of CRC are, in most cases, asymptomatic; therefore, numerous screening programs are focused on the early detection of CRC [110]. Strategies for CRC screening programs differ in terms of the age at which screening begins, frequency of screening and methods used. These parameters can be optimised to increase the sensitivity and efficiency of screening. The prognosis of patients with CRC also seems to depend on the length of the diagnostic interval between the appearance of the first symptoms and CRC diagnosis [121], as well as the morphological types of polyps presented in the colorectum [122].

Current screening methods have several limitations due to their low specificity and sensitivity, invasiveness, or high cost. There are several promising markers, including detection of mutations, proteins and mRNA, which are commercially available or in the testing stage [106,110]. In clinical practice, there are few miRNA-based diagnostic tools [4]. Diagnostic biomarkers for CRC could be classified according to the examined specimen into tissue, circulation and faecal markers [106,110].

Colonoscopy and sigmoidoscopy are commonly used invasive methods to detect the presence of polyps in the colorectum and to resect samples for biopsy. Several genetic tests for evaluating the diagnosis and prognosis of CRC from tissue samples beside histopathological and immunochemical analyses are currently used in clinical practice [106,110,123]. The microsatellite instability (MSI) test is based on a measurement of a panel of several markers including BAT25, BAT26, D5S346, D2S123 and D17S250. The indicator of MSI is the presence of mutation in genes involved in mismatch repair, e.g., hMLH1 and hMSH2 [110,123]. Other gene mutations showing a strong effect on CRC development include mutations in the genes KRAS, BRAF and APC [110].

Recently, circulation markers, e.g., glycoproteins CEA (carcinoembryonic antigen) and CA19-9 (cancer antigen 19-9) are widely used for CRC diagnosis [124,125,126] but miRNAs in the circulation are also promising markers for CRC diagnosis. miRNAs in vesicles were reported to be resistant to degradation by RNase A, and their levels stayed stable during 24 h of incubation at room temperature or during several freeze–thaw cycles [127,128].

A panel of several dysregulated miRNAs in the circulation seems to be a more efficient diagnostic and prognostic factor than a single miRNA. Previous studies showed that a panel of selected miRNAs showed higher sensitivity as a prognostic marker for early CRC diagnosis, compared with CEA. However, the highest sensitivity was displayed when a combination of CEA and a miRNA panel were used together [98,102,124,129].

The up-regulated miRNA clusters selected for our study also showed associations with clinicopathological characteristics, such as TNM stage, tumour size, metastasis and low differentiation. The presence of distant or lymph node metastasis predominantly positively correlated with up-regulation of all miRNAs from clusters miR-17/92a-1, miR-183/96/182, miR-191/425 and miR-200c/141. Advanced TNM stage was associated with increased expression of several miRNAs from clusters miR-17/92a-1, miR-106a/93/25, miR-183/96/182 and miR-200c/141 in serum or tumour tissue (Appendix A).

miRNAs are a promising prognostic factor for predicting patient survival and disease recurrence. Up-regulation of the majority of miRNAs from clusters miR-17/92a-1, miR-106b/363, miR-181-1/182-1, miR-181-2/182-2 and miR-183/96/182 in tumour tissue or the circulation was associated with shorter patient survival and increased disease recurrence. The expression of both members of the miR-222/221 cluster positively correlated with disease recurrence. miRNAs from cluster miR-106b/93/25 were differentially associated with patient prognosis. Increased expression of miR-106b and miR-93 in tumour tissue was associated with improved survival and a lower incidence of relapse, whereas up-regulation of miR-25 in tumour tissue was associated with worse survival (Table 5).

### 3.7. Association of miRNA Cluster Expression with Response to Chemotherapy

The development of colorectal tumours is usually a slow, multi-step process accompanied by histological, morphological, and genetic changes that accumulate over time. Colorectal tumours develop from benign precancerous polyps, predominantly from adenomas and sessile serrated polyps [199]. The most common genetic changes associated with the early stages of CRC development are chromosomal and microsatellite instabilities, which are associated with drug resistance. Chromosomal instability results mostly from defects in chromosomal segregation, decreased telomere stability and insufficient DNA damage repair [200]. Microsatellites are short DNA motifs which are distributed throughout the genome. Repetitive microsatellite DNA sequences are more resistant to errors in replication, thus the accumulation of mistakes in microsatellites points to a deficit in the mis-match repair system (MMR), which rectifies mismatches in the DNA sequence during replication [201]. Chromosomal and microsatellite instabilities are closely related to an increased number of gene mutations [200]. Within the selected up-regulated clusters, the expression of miR-221, miR-224, miR-181b and miR-92 in CRC tissue was associated with microsatellite instability [198,202]. Mutations of key regulatory genes such as thymidylate synthase or components of the p53 pathway may also lead to the development of resistance to chemotherapy [108,203].

The most commonly used chemotherapeutics for CRC treatment are 5-fluorouracil (5-FU) and oxaliplatin. 5-FU is a uracil analogue, which, after intracellular metabolization to several active metabolites, inhibits RNA synthesis [204]. Platinum-containing drugs, e.g., oxaliplatin, cisplatin and carboplatin, bind preferentially to DNA molecules and inhibit DNA replication and transcription through the formation of DNA adducts [205].

Several miRNAs from up-regulated clusters are involved in signalling pathways associated with sensitivity to chemotherapy and have the potential to become a tool in CRC treatment [206]. Higher expression of miR-27a, miR-20b and miR-106a-5p in CRC tissue was associated with a worse response to treatment with 5-FU [207,208,209]. The expression of miR-17-5p, miR-19b, miR-20a and miR-93 from the miR-17/92a-1 cluster and its paralogous cluster miR-106b/93/25 was increased in CRC tumours resistant to 5-FU-based adjuvant therapy compared with chemo-sensitive tumours. Increased expression of miR-17-3p in the tumour tissues of non-responsive patients was correlated with shorter survival [150]. Additionally, high expression of miR-92a-3p in tumour tissues was associated with worse response to neoadjuvant therapy based on 5-FU [210].

Two miRNAs from the miR-17/92a-1 cluster, miR-20a [211] and miR-19a [212] were determined as serum biomarkers for non-responsiveness to FOLFOX therapy (combined oxaliplatin and 5-FU treatment). High expression of miR-17-3p and miR-106a in the serum of patients with stage II and III tumours treated with adjuvant chemotherapy was associated with worse therapeutic outcomes [104]. Increased expression of miR-96-5p in serum exosomes was revealed as a potential predictive biomarker for chemoresistance [213]. miR-106a and miR-130b were validated as plasma biomarkers for the response to 5-FU/oxaliplatin therapy, and increased expression of these miRNAs was associated with a worse response to therapy [214]. Treatment of CRC patients with 5-FU chemotherapy led to increased expression of miR-23a-3p in tumour tissues, compared with untreated tumour tissues [215].

### 3.8. Target Genes of miRNA Clusters

miRNA functions in CRC development are mediated via target genes and key pathways regulating cell proliferation and migration, such as the Wnt/β-catenin, PI3K/Akt/mTOR, TGF-ß (transforming growth factor beta) and EGFR (epidermal growth factor receptor) signalling pathways [216,217]. Deregulated miRNAs in CRC tissue could be classified as oncogenic miRNAs (oncomiRs), promoting tumour development, metastasis or resistance to therapy or tumour-suppressive miRNAs with the opposite effects [218].

Based on recent experimental evidence from studies using CRC cell lines, 181 genes were confirmed as target genes of miRNAs from selected up-regulated clusters. Approximately one sixth of these genes (33) were targeted by two or more miRNAs. An oncogenic or tumour suppressive role in CRC was attributed to target genes based on experimental evidence (Table 6 and Appendix A).

Next, the functional relationship between the expression of miRNAs from selected up-regulated clusters and their target genes was analysed. The most frequently targeted oncogene (four times) was catenin beta 1 (CTNNB1). The most targeted tumour suppressors were phosphatase and tensin homolog (PTEN) and gamma-amino-butyric acid type B receptor 1 (GABBR1), inhibited by eight and five miRNAs, respectively. The genes SMAD7, TGFBR2 and TNFAIP3 were associated with both tumour suppressor and oncogenic roles. The target genes of miRNAs from selected clusters are listed in Table 6 and Appendix A.

For complex classification of all target genes of selected miRNAs, we used GO-Slim analysis performed by the Panther Classification System [12]. Genes were classified according to molecular function, biological processes, protein class and pathway. An overrepresentation test was used for each classification to reveal significantly enriched categories (Appendix A).

Classification according to GO molecular function revealed that 128 target genes could be attributed to one of the GO molecular function categories, while 54 target genes fell into the category unclassified. The most overrepresented category of GO molecular function was binding (GO:0005488), with 87 target genes, and this category was the only one showing significant enrichment in the overrepresentation test (*p* < 0.001; FDR < 0.001). The results of GO-Slim molecular function analysis are provided in Figure 2.

Analysis according to GO-Slim biological process classified the target genes into 17 categories. The overrepresentation test showed seven significantly enriched categories (*p* < 0.05; FDR < 0.05), and 45 target genes remained in the category unclassified according to the biological process terms. The biological processes categories identified according to GO-Slim analysis are depicted in Figure 3.

Analysis according to Panther protein class showed that the target genes of miRNA clusters up-regulated in CRC belonged to 19 categories, and 52 target genes fell into the category unclassified. Classification according to the protein class overrepresentation test revealed that the significantly enriched categories were protein modifying enzyme and gene-specific transcriptional regulator (*p* < 0.01; FDR < 0.01). Figure 4 depicts all categories according to Panther protein class classification.

Analysis according to Panther pathways [12] showed that target genes were involved in 68 pathways, and 106 target genes were in the category unclassified according to Panther pathways terms. The overrepresentation test revealed 26 significantly enriched pathways. Figure 5 depicts only significantly enriched pathways according to the Panther pathways classification.

The pathway with the highest number of included genes was cholecystokinin receptor (CCKR) signalling map (P06959), which included 17 genes from our analysis. The gastrointestinal peptide hormones gastrin and cholecystokinin (CCK) are ligands of two G-protein coupled receptors: cholecystokinin 1 receptor (CCK1R) and cholecystokinin 2 receptor (CCK2R). CCK1R and CCK2R are involved in the regulation of cell proliferation and migration through several downstream signalling pathways [219]. Nine analysed genes enriched in the CCKR signalling map pathway, PTEN, FOXO1, FOXO3, KLF4, GSK3B, MYC, CCND1, TCF4 and CTNNB1, were targeted by multiple miRNAs from different families (miRNAs and their target genes are listed in Table 6 and Appendix A).

The CCKR signalling map pathway was associated with several tumour suppressors. Among them, the gene PTEN is regulated by eight miRNAs from eight clusters, and the pro-apoptotic transcription factors Forkhead Box O3 (FOXO3) and Forkhead Box O1 (FOXO1), associated with the inhibition of cell proliferation and promotion of apoptosis, are targeted by miR-96-5p and miR-182-5p from the miR-182/96/183 cluster [62,220]. The zinc-finger transcription factor Krüppel-like factor 4 (KLF4), which inhibits cell proliferation and migration, is targeted by miR-92a-3p [210] and miR-29a-3p [190]. Serine-threonine kinase glycogen synthase kinase 3 beta (GSK3B) is targeted by miR-224 [221] and miR-452-3p [195] from cluster miR-452/miR-224. Down-regulation of GSK3B leads to increased cell proliferation and migration [195].

On the other hand, miR-93-5p and miR-182-5p suppress tumorigenesis in vivo and cell growth through targeting of the transcription factor MYC [222,223]. Decreased expression of CCND1 (cyclin D1) caused by the inhibitory effect of miR-93-5p, miR-18a-5p and miR-96-5p leads to decreased cell proliferation and cell cycle arrest [167,222,224]. The transcription factor TCF4 (transcription factor 4) and cell adhesion protein CTNNB1 promote cell proliferation and resistance to 5-FU [56]. TCF4 and CTNNB1 are targeted by miR-181a-5p [56] and miR-203a-3p [225,226]. CTNNB1 is also a target gene of miR-93-5p [222].

The gonadotropin-releasing hormone receptor pathway (P06664) was associated with 16 target genes of up-regulated miRNA clusters. GSK3B, SMAD4 and CTNNB1 were the most frequently targeted by miRNAs from several families. Signal transduction protein SMAD4, a member of the SMAD family, is a suppressor of proliferation and increases sensitivity to oxaliplatin [227]. SMAD4 is targeted by miR-20a-5p [155], miR-19b-3p [227] and miR-18a-5p [228] that are members of cluster miR-17/92a-1.

Thirteen genes targeted by miRNA clusters up-regulated in CRC tissue were associated with the angiogenesis (P00005) pathway. The genes KRAS, GSK3B, HIF1A and CTNNB1 were targeted by miRNAs from different families. KRAS (KRAS Proto-Oncogene, GTPase) is an oncogene frequently activated by mutation in CRC [110]. Inhibition of KRAS expression by miR-19a-3p [229] and miR-96-5p [167] leads to decreased cell growth and down-regulation of vascular endothelial growth factor A (VEGFA). Similarly, reduced expression of angiogenesis-inducing factor hypoxia induced factor-1α (HIF1A) by targeting with miR-18a-5p [230] and miR-93-5p [81] leads to decreased cell growth and tumour angiogenesis inhibition in vivo.

Transcription factor p53 plays an important role in maintaining DNA integrity and regulation of the cell cycle [231]. The p53 pathway (P00059) was associated with 11 target genes; among them, the genes ATM, CDKN1A, SIRT1, THBS1 and PTEN were targeted by miRNAs from multiple families. Reduced expression of the DNA-damage repair protein ATM (ataxia telangiectasia mutated) mediated by miR-18a-5p [153] and miR-203a-3p [232] leads to oxaliplatin resistance and decreased DNA repair. The kinase cyclin dependent kinase inhibitor 1A CDKN1A), also known as p21, is an important negative regulator of the cell cycle. CDKN1A is targeted by miR-224-5p [233], miR-106b [234] and miR-20a-5p [235]. Reduced expression of extracellular matrix protein thrombospondin 1 (THBS1) via interaction with miR-19a-3p [236], miR-182-5p [237] and miR-203a-3p [238] promotes proliferation and migration. Increased expression of sirtuin 1 (SIRT1) promotes cell proliferation and oxaliplatin resistance, and these effects can be reversed by targeting with miR-141-3p [78] and miR-29b-3p [239].

Oncogenic mutations of components of the Wnt signalling pathway, mostly mutations in APC (the APC regulator of WNT signalling pathway), are key factors in CRC development [231]. The Wnt signalling pathway (P00057) is associated with 11 target genes. CCND1, GSK3B, SMAD4, CTNNB1 and MYC are targeted by miRNAs with the use of different family-specific seed sequences.

Panther pathway analysis was also performed for individual miRNA clusters. Table 7 lists pathways that were significantly increased in the statistical overrepresentation test according to target genes of particular clusters. In this analysis, we only included clusters for which there were enough data about target genes (seven clusters), and clusters miR-181c/181d, miR-191/425, miR-221/222, miR-23a/27a/24, miR-29b/29a, and miR-452/224 had to be omitted.

The pathways CCKR signalling map (P06959) and p53 pathway feedback loops 2 (P04398) were overrepresented in the target genes of five miRNA clusters each. The highest number of overrepresented pathways were associated with clusters miR-17/92a-1 (13 pathways) and miR-106b/93/25 (5 pathways).

## 4. Discussion

An increasing number of studies refer to the important role of miRNA dysregulation in colorectal cancer. Many usually up-regulated miRNAs have also been suggested as promising biomarkers for CRC diagnosis or prognosis assessment [2,3]. Therefore, the main aim of this systematic review was to provide selection and functional analysis of miRNA clusters up-regulated in CRC tissue.

Based on data from available experimental research studies, 15 miRNA clusters with up-regulated expression in CRC were selected. Approximately half of these clusters are embedded in intronic areas of genes, and the rest are located in intergenic regions. Many transcription factors are involved in the regulation of miRNA clusters, and among them, MYC, ZEB1 and p53 play important roles in CRC.

Increased expression of ZEB1 was associated with worse survival in several types of digestive cancers, e.g., pancreatic, gastric and colorectal cancers [240]. The transcription factor ZEB1 inhibits the expression of miRNA clusters miR-200c/141 [241], miR-183/96/182 and miR-203a/203b [242].

The transcription factor MYC seems to play an important role in CRC development through interaction with several protein coding genes and non-coding RNAs [243,244]. Expression of up-regulated miR-181d [34] and the clusters miR-29b-1/29a [245], miR-17/92a-1 [33,246] is also under the control of MYC in CRC. MYC shows miRNA specific influence as expression of miR-17/92a-1 and miR-181d is induced and expression miR-29b-1/29a is down-regulated by this transcription factor. While miR-17/92a-1 and miR-181d are positively associated with advanced TNM stage and the presence of metastases, information about cluster miR-29b-1/29a is rather inconclusive in this respect.

The induction of miR-17/92a-1 by MYC is in good agreement with the observation that MYC expression is up-regulated in CRC tissue compared to normal tissue and it is associated with worse survival and chemoresistance. Moreover, inhibition of MYC expression led to increased apoptosis in vitro [247,248]. On the other, increased expression of MYC in tumour tissue was also associated with improved survival and smaller tumour size and better survival of patients with CRC [249]. As there is no straightforward explanation for these opposing observations, it seems that more research aimed at exploring the details of the biological and clinical context of measured samples is needed to reveal roles of MYC in CRC progression.

The transcription factor p53 shows a significant impact on the expression of many miRNAs from selected up-regulated clusters. In particular, p53 inhibits expression of all members of the cluster miR-17/92a-1, namely miR-27a, miR-29a, miR-222 and miR-224. On the other hand, p53 induces the expression of miR-106a, miR-221 and the whole miR-200c/141 cluster. In general, p53 shows inhibitory rather than stimulatory effects with respect to up-regulated clusters in CRC [250,251].

It is no surprise that members of the cluster miR-17/92a-1 are convincingly associated with metastasis occurrence and poor survival in CRC patients. Members of this intronic cluster have been shown to exert a high level of co-expression amongst themselves as well as with the host gene. However, there are exceptions to this observation that can issue from the presence of binding sites of 138 transcription factors including p53, CREB1 (cAMP responsive element binding protein 1), MYC and SMAD1 (SMAD family member 1) [250,251]. According to Panther pathways analysis, this cluster employs 12 pathways involved in regulation of cancer progression, including p53, Ras, HIF, VEGF and TGFβ signalling pathways and some others. Within the selection of up-regulated clusters, the miR-17/92a-1 cluster shows the strongest association with poor patient outcomes, it is the most studied cluster, and its effects are realised by the highest number of regulatory pathways.

On the other hand, the clusters miR-106b/93/25 and miR-183/96/182 also show strong associations with metastases occurrence in CRC patients (although the association with survival is weaker, compared with that of miR-17/92a-1). miR-106b/93/25 is, similarly to miR-17/92a-1, an intronic cluster showing partial co-expression with the host gene and the potential to respond to up to 148 transcription factors, including MYC, RB1 (RB transcription corepressor 1), EGR1 (early growth response 1) and CREB1 [250,251]. Among others, miR-106b/93/25 utilises the Wnt, CCKR and p53 signalling pathways. Expression of the miR-183/96/182 cluster is regulated by 42 transcription factors, and as it is located in intergenic locus in the genome, it possesses its own TSS. Only the PI3 kinase pathway was significantly linked to miR-183/96/182, according to Panther pathway analysis.

Finally, high expression of miR-106a/363 showed a strong association with poor survival of CRC patients. Expression of miR-106a/363 seems to be regulated by 25 transcription factors [250,251], and miR-363 is likely to be regulated in a different manner to the rest of the cluster, as it frequently shows a different response to treatment and differs in correlation studies (Table 5, Appendix A).

Clusters showing the strongest association with metastasis occurrence and/or poor survival (miR-17/92a-1, miR-106b/93/25, miR-183/96/182 and miR-106a/363) share nine regulatory domains: EP300 (E1A binding protein p300), ERG (ETS transcription factor ERG), MYC, GTF2I (General transcription factor IIi), MAX (MYC associated factor X), AR (Androgen receptor), ETS1 (ETS proto-oncogene 1), MYCN (MYCN Proto-Oncogene) and TFAP2A (Transcription factor AP-2 alpha) [250,251].

On the other hand, at least some members of the clusters miR-23a/27a/24-2, miR-203a/203b and miR-200c/141 show tumour-suppressive functions, which is in accordance with the number of their target genes with oncogenic character. However, this regulatory relationship was not reflected in a positive correlation between high expression of miR-23a/27a/24-2, miR-203a/203b and miR-200c/141 cluster members and patient survival. According to Panther pathways analysis, cluster miR-203a/203b exerts it functions via apoptosis, CCKR and the p53 signal pathway, whereas miR-200c/141 is related only to the GnRH receptor signalling pathway. The abovementioned clusters share 18 common regulatory domains: bromodomain containing 4 (BRD4), lysine demethylase 5B (KDM5B), mediator complex subunit 1 (MED1), E1A binding protein p300 (EP300), transcription factor AP-2 gamma (TFAP2C), Kruppel-like factor 5 (KLF5), ERG, MYC, MAX, E2F transcription factor 1 (E2F1), AR, aryl hydrocarbon receptor nuclear translocator like (ARNTL), HIF1A, EGR1, nuclear receptor subfamily 2 group F member 2 (NR2F2), Sp1 transcription factor (SP1), transcription factor AP-4 (TFAP4) and oestrogen receptor 1 (ESR1) [250,251].

Surprisingly, clusters with the strongest oncogenic and tumour suppressor potential share five regulatory regions (MAX, ERG, EP300, AR and MYC). The binding domains specific for oncogenic clusters are GTF2I, ETS1, MYCN and TFAP2A. Three of them, ETS1, MYCN and TFAP2A, have been previously associated with the cancer progression and all of them show tumour promoting effects [252,253,254].

The miRNA clusters listed in this study predominantly have increased expression in CRC tissue according to the analysed studies. The majority of research studies also confirmed up-regulation of all miRNAs from clusters miR-17/92a-1, miR-181a-1/181b-1, miR-181a-2/181b-2, miR-181c/181d, miR-183/96/182, miR-191/425, miR-200c/141, miR-203a/203b, miR-222/221 and miR-301b/130b in the circulation of patients with CRC (Appendix A). On the other hand, in stool samples of CRC patients, only clusters miR-17/92a-1 and miR-183/96/182 exerted increased values (Table 4). This inconsistency can be attributed to many factors, including the manner in which colorectal cancer influences food intake and intestinal transition time in individual patients.

Occasionally, one or two studies were not in line with the majority of reports concerning up- or down-regulation of particular miRNA levels in tissues and/or the circulation. The reason for this may issue from the existence of several variants of mature miRNA known as isomirs [255]. Isomirs differ slightly in length due to impaired cleavage by Drosha and Dicer [256] or because of post-transcriptional modifications [255]. Rhythmic changes in the expression of several miRNAs were observed in mammalian tissues [257,258,259] or the circulation [260]; therefore, we assume that rhythmic expression may also increase the variability in miRNA expression measurement when sampling is not performed at the same time of day.

The expression of miRNA in tumour tissue, the circulation and stool samples was correlated with clinicopathological characteristics, survival, disease recurrence and response to chemotherapy. Up-regulated expression of clusters miR-106b/93/25, miR-17-92a-1 and miR-183/96/182 was strongly associated with the presence of distant or lymph node metastasis (Appendix A). Worse survival of patients was convincingly associated with increased expression of clusters miR-17/92a-1 and miR-106a/363 (Table 5).

miRNAs involved in clusters with up-regulated expression in CRC tissue are frequently associated with the promotion of tumour development and show oncogenic effects. However, our study revealed exceptions to this pattern. Several up-regulated miRNAs were referred to as tumour-suppressors in studies performed under in vitro conditions with the use of CRC cell lines. The cluster miR-23a/27a/24-2 was predominantly up-regulated in CRC tissue and the circulation, but miR-24-2 was also negatively correlated with advanced tumour stage and metastasis. Increased expression of miR-24 in CRC was associated mostly with a tumour-suppressor effect through the targeting of genes with oncogenic potential. This indicates that miR-24 may have a different role in CRC development from other members of the cluster, namely miR-27a and miR-23a. Similarly, members of the cluster miR-200c/141 show predominantly tumour-suppressing effects in CRC cells through targeting of oncogenes (Table 6 and Appendix A). According to these facts, we assume that the cluster miR-200c/141 could be classified as a tumour-suppressive rather than an oncogenic cluster. Target genes of the miRNA cluster miR-203a/203b in CRC cell lines were also mainly oncogenes, which implies a tumour-suppressive role of this cluster. However, the expression of miR-203a and miR-203b was predominantly increased in CRC tissue and the circulation.

GO-Slim and Panther pathway analyses of target genes were performed to better understand the regulatory role of up-regulated miRNA clusters and their target genes in CRC. Based on information from experimental studies selected according to the defined selection criteria, 181 genes were identified as experimentally proven target genes of miRNAs from clusters up-regulated in CRC tissue. The ratio of tumour suppressors and oncogenes was nearly equal, as there were 86 oncogenes and 92 tumour suppressors (and 3 genes had no clear role in oncogenesis).

GO-SLIM classification of target genes of up-regulated clusters in CRC according to class molecular function revealed that the overrepresented category was binding (GO:0005488), which is in accordance with previous studies [261,262].

Analysis of target genes according to GO-Slim biological process revealed seven significantly overrepresented categories: biological regulation (GO:0065007), cellular process (GO:0009987), developmental process (GO:0032502), locomotion (GO:0040011), meta-bolic process (GO:0008152), response to stimulus (GO:0050896) and signalling (GO:0023052).

Classification according to Panther protein class showed that target genes were included in 19 categories. Significantly overrepresented categories were protein modifying enzyme (PC00260) and gene-specific transcriptional regulator (PC00264).

Analysis of target genes according to Panther pathways categories revealed 26 significantly overrepresented pathways. A majority of target genes were associated with pathways CCKR signalling map (P06959), gonadotropin-releasing hormone receptor pathway (P06664), angiogenesis (P00005), Wnt signalling pathway (P00057) and p53 pathway (P00059). A previous study analysing target genes of miRNAs deregulated in CRC tissue revealed that among the 10 most enriched pathways were angiogenesis (P00005), the Wnt signalling pathway (P00057) and the p53 pathway (P00059) [263].

When Panther pathway analysis was performed separately for tumour suppressors and oncogenes, interestingly, biological process rhythmic process (GO:0048511) and pathway circadian clock system (P00015) were linked only to tumour-suppressor genes (CRY2 and FBXL3). On the other hand, the pathways cell cycle (P00013) and VEGF signalling pathway (P00056) were linked only to oncogenes (KRAS, ETS1, VEGFA, PRKCZ, HIF1A and SPHK2). Processes exclusively linked to tumour-suppressive genes were associated with cluster miR-181c/181d, whereas pathways associated only with oncogenes were used by the clusters miR-17/92a-1, miR-182/96/183, miR-106b/25/93, miR-106a/363 and miR-200c/141.

In summary, this review focused on selecting and analysing miRNA clusters with up-regulated expression in CRC. miRNAs are an intensively studied class of small non-coding RNAs with promising biomarker potential not only in tumour tissue but also in the circulation. To increase the specificity of miRNA-based diagnostics, there is a strong tendency to use panels of several miRNAs rather than a single miRNA [217,264,265].

Thanks to prominent advances in in vivo delivery systems, animal and clinical studies employing miRNA administration were strongly facilitated. This effort was followed by the release of several miRNA-based therapeutics that are being clinically tested or are already being used in the treatment of a wide spectrum of diseases including cancer malformations. miR-29b-1-5p and miR-222-3p, mentioned in this review, are part of miRNA panel for diagnostic thyroid and pancreatic cancer. miR-29b is used for the treatment of fibrosis, and miR-92 turned out to be useful for treatment of ischemia [4]. However, many more miRNAs or miRNA inhibitors have been clinically tested, as reviewed elsewhere [266].

The effort of pharmaceutical companies focused on the clinical use of miRNA was reflected by thousands of patent applications in US and European patent databases. The highest number of registered patents is related to oncological diseases including colon cancer. There are several commercial companies developing miRNA-based therapeutics focused on the clinical use of miRNA. Among them, tumour-suppressor miR-34a was tested as a tool for the treatment of colon cancer. Similarly, the delivery of miR-145 and miR-33a has been tested for in vivo treatment of colon cancer [267].

miRNAs located in clusters are likely to be transcribed as a single polycistronic transcript affected by same transcription factors and epigenetic changes [268,269]. This feature of clustered miRNAs has not recently been involved in strategies for new drug development; however, we believe that it has strong potential to increase the efficiency of miRNA-based treatment. Considering whole clusters instead of single miRNAs can be useful, especially when whole clusters show oncostatic or tumour-promoting potential.

In this review, we also analysed interactions among miRNAs and long non-coding RNAs. lncRNAs are frequently host genes of miRNAs, and they show the capacity to inhibit miRNA effects through sponging them. Better understanding of miRNA transcriptional and post-transcriptional regulation can further facilitate development of miRNA-based therapeutics tested and/or used in cancer treatment.

## 5. Conclusions

Our results indicate that members of clusters with up-regulated expression in CRC frequently show oncogenic potential. However, the extent of the tumour-promoting effect varies greatly among them. Overall, it is possible to conclude that clusters with up-regulated expression seem to be associated to a greater extent with worse patient prognosis, although exceptions for particular miRNAs have been reported. Similarly, when the influence of miRNAs from selected up-regulated clusters on chemotherapy is reported, it is usually in the induction of chemoresistance rather than chemosensitivity. To use up-regulated miRNAs as CRC markers would require the assumption that levels of miRNAs in cancer tissue, circulation and faeces correlate. This prerequisite is not met for all up-regulated clusters; however, it is valid for miR-17/92a-1 and miR-183/96/182. The correlation between miRNA levels in the tumour tissue and the circulation is more significant than that reported in tumour tissue and faeces. The location of up-regulated clusters is variable; however, no cluster was located in an exon. In spite of huge progress on the roles of miRNA in CRC progression, more effort to reveal the mechanisms of their action is needed, as not all members of selected up-regulated clusters could be linked to a particular intracellular pathway. Based on the abovementioned conclusions, it seems that miRNA from up-regulated clusters in CRC are promising targets for future translational research.

## Figures and Tables

**Figure 1 cancers-13-02979-f001:**
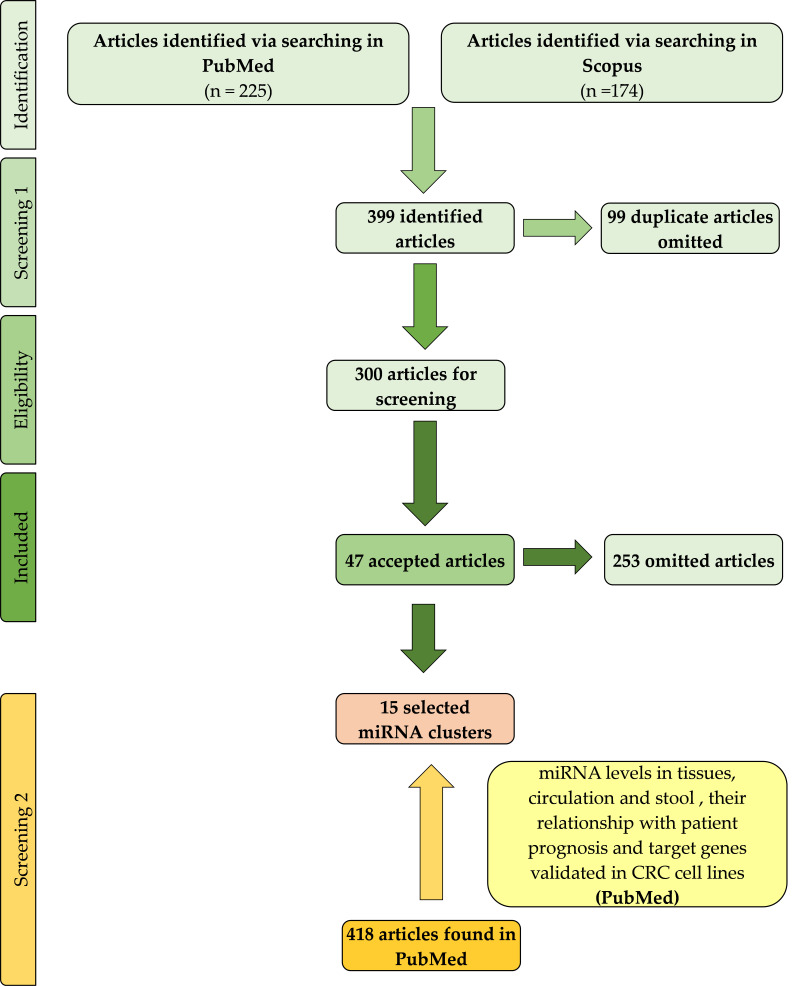
Study flow diagram of selection for up-regulated miRNA clusters.

**Figure 2 cancers-13-02979-f002:**
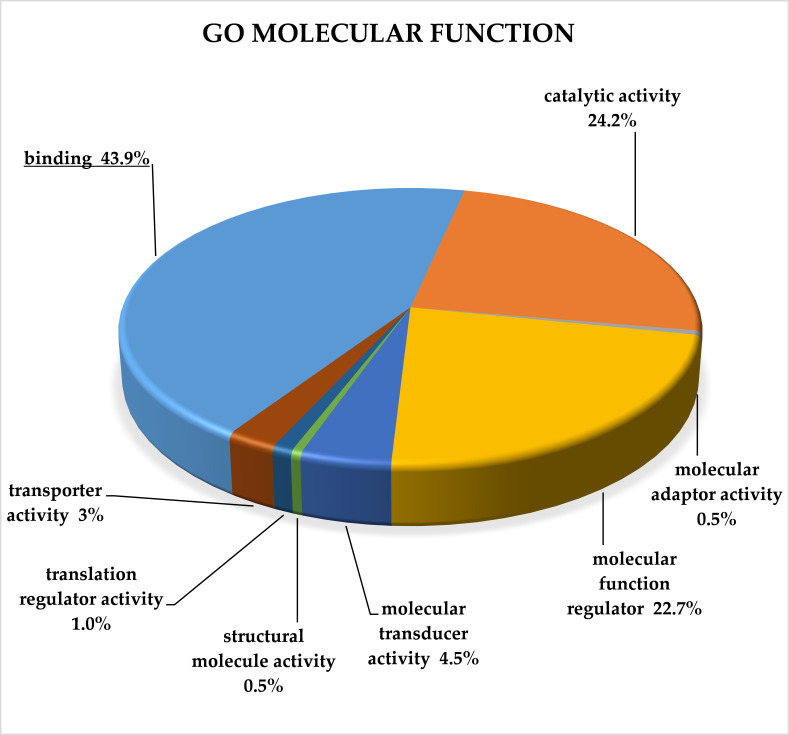
GO-Slim analysis of target genes of up-regulated miRNAs in CRC. Classification is according to molecular function terms, and all terms are plotted. The significantly overrepresented term is underlined (*p* < 0.001; FDR < 0.001). FDR—false discovery rate.

**Figure 3 cancers-13-02979-f003:**
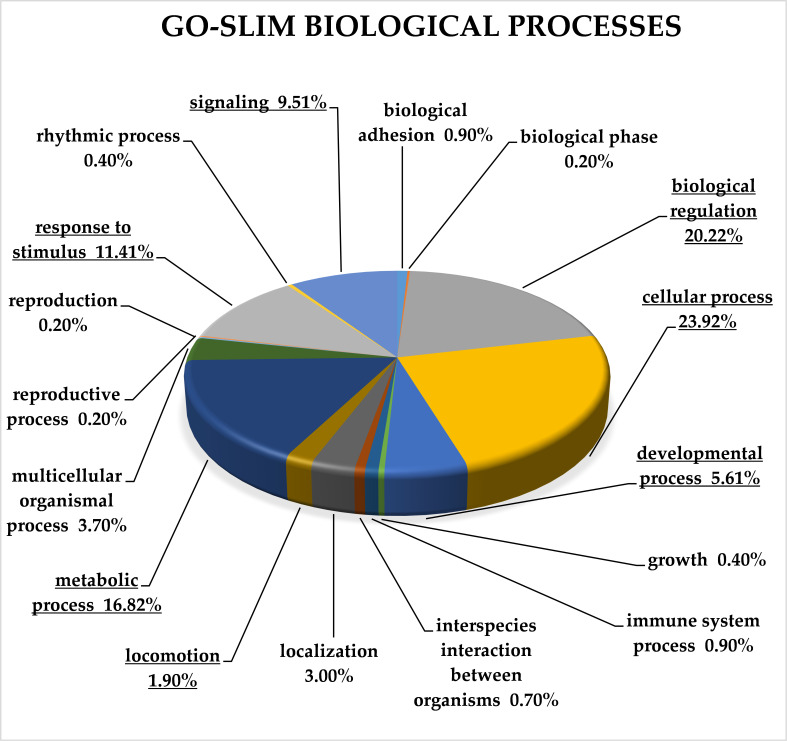
GO-Slim analysis of target genes of up-regulated miRNAs in CRC. Classification is according to biological processes terms. Significantly overrepresented terms are underlined (*p* < 0.05; FDR < 0.05). FDR—false discovery rate.

**Figure 4 cancers-13-02979-f004:**
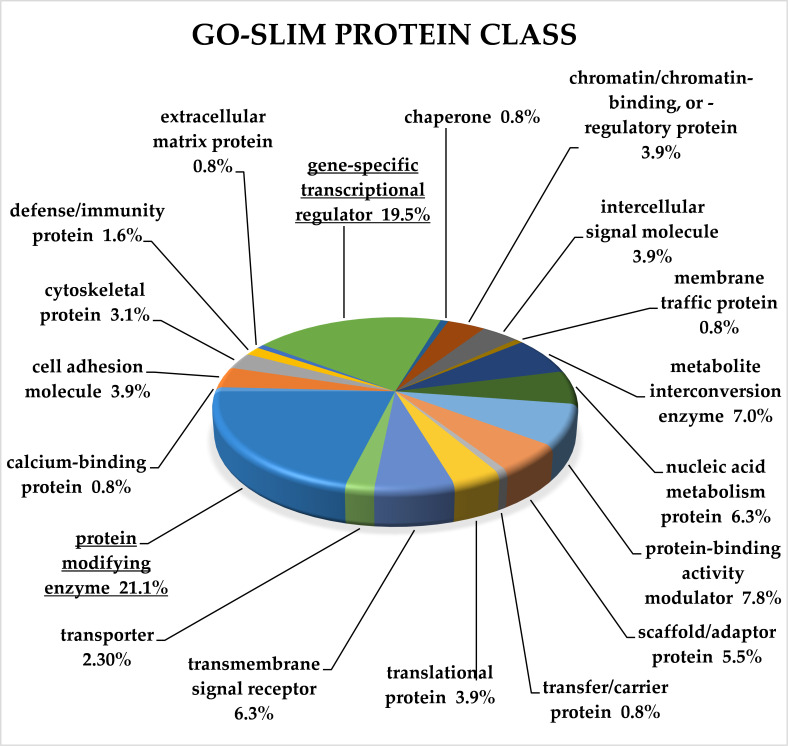
GO-Slim analysis of target genes of up-regulated miRNAs in CRC. Classification is ac-cording to protein class terms, and all terms are plotted. Significantly overrepresented terms are underlined (*p* < 0.01; FDR < 0.01). FDR—false discovery rate.

**Figure 5 cancers-13-02979-f005:**
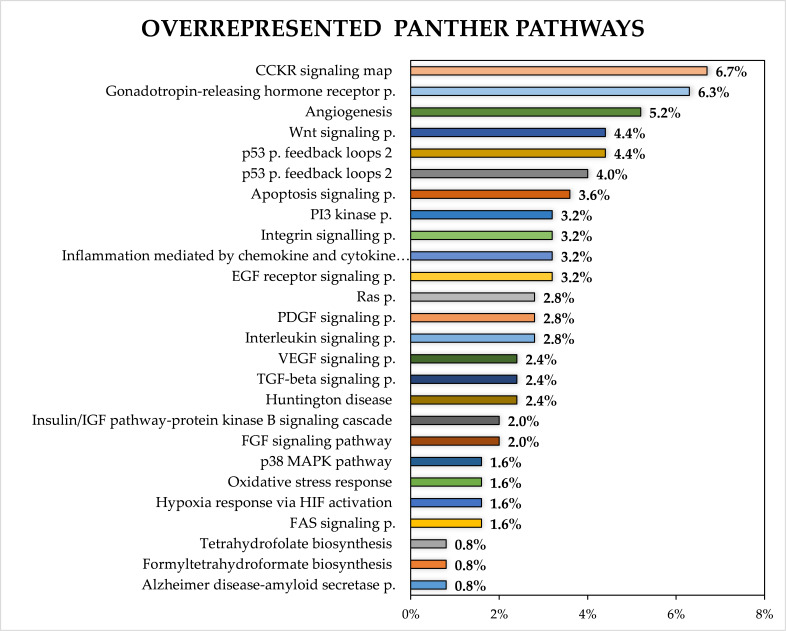
GO-slim analysis of target genes of up-regulated miRNAs in CRC. Classification is according to Panther pathways terms, and only significantly overrepresented terms are plotted (*p* < 0.05; FDR < 0.05). FDR—false discovery rate.

**Table 1 cancers-13-02979-t001:** Host gene and genomic location of miRNA clusters.

Cluster Chromosome	Host Gene/*Locus* Name	Host Gene Type	Cluster Location	Expression Regulation and Transcription Site (TSS)
miR-106a/18b/20b/19b-2/92a-2 *chr. X*	*Xq26.2* [15]	Non-coding	intergenic	Own TSS [10]
miR-106b/93/25 *chr. 7*	MCM7 [15]	Protein coding	intron	Co-expression of miR-25 with the host gene [16]. Host gene TSS [17,18,19]
miR-17/18a/19a/20a/19b-1/92a-1 *chr.13*	MIR17HG [15]	Long non-coding	intron	Co-expression of miR-17, miR-18a, miR-20a and miR-92a with the host gene [16,20]. Multiple TSS [10,11]
miR-181a-1/181b-1 *chr. 1*	MIR181A1HG [21]	Long non-coding	intron	Multiple TSS [10]
miR-181a-2/181b-2 *chr. 9*	NR6A1 [22]	Protein coding	intron	Multiple TSS [10]
miR-181c/181d *chr. 19*	NANOS3 [22,23,24]	Protein coding	intron	Own TSS [10]
miR-183/96/182 *chr. 7*	MIR96 [23,24]	Non-coding	intergenic	Own TSS [10]
miR-191/425 *chr. 3*	NDUFA3 [16]	Protein coding	intron	Host gene TSS [25]
miR-200c/141 *chr. 12*	MIR200CHG [26]	Long non-coding	intergenic	Own TSS [27]
miR-203a/203b *chr. 14*	MIR203A [28]	Non-coding	intergenic	Own TSS [10]
miR-222/221 *chr. X*	MIR222HG [25]	Long non-coding	intergenic	Own TSS [10]
miR-23a/27a/24-2 *chr. 19*	MIR23AHG [26]	Long non-coding	intergenic	Own TSS [29]
miR-29b-1/29a *chr. 7*	LINC-PINT [30]	Long non-coding	intergenic	Multiple TSS [10]
miR-301b/miR-130b *chr. 22*	PPIL2 [31]	Protein coding	intron	Multiple TSS [10]
miR-452/224 *chr. X*	GABRE [9]	Protein coding	intron	Co-expression with the host gene [17,18,19]

Abbreviations—MCM7 (minichromosome maintenance complex component 7), MIR17HG (miR-17-92a-1 cluster host gene), MIR181A1HG (MIR181A1 host gene), NR6A1 (nuclear receptor subfamily 6 group A member 1), NANOS3 (nanos C2HC-type zinc finger 3), NDUFA3 (NADH:ubiquinone oxidoreductase subunit A3), MIR200CHG (MIR200C and MIR141 host gene), MIR222HG (miR222/221 cluster host gene), MIR23AHG (miR-23a/27a/24-2 cluster host gene), LINC-PINT (long intergenic non-protein coding RNA, p53 induced transcript), PPIL2 (peptidylprolyl isomerase like 2), GABRE (gamma-aminobutyric acid type A receptor subunit epsilon). TSS—transcription start site.

**Table 2 cancers-13-02979-t002:** List of lncRNAs and circRNAs sponging up-regulated miRNA clusters in human CRC cell lines.

miRNA Cluster	Sponging ncRNA	ncRNA Class	Reference
miR-183/96/**182**	AGER-1	intronic ncRNA	[48]
**miR-203a**/203b	BANCR	lincRNA	[49]
miR-106b/**93**/25	CA3-AS1	antisense RNA	[50]
**miR-181a-1/181b-1** **miR-181a-2/181b-2**	CCAT1	lincRNA	[41,42]
**miR-106b**/93/25	circ_000984 (CDK6)	circRNA	[51]
**miR-183**/96/182	circ_0026344 (ACVRL1)	circRNA	[52]
**miR-106b**/93/25	circ_0055625 (DUSP2)	circRNA	[53]
**miR-203a**/203b	circ_0079993 (POLR2J4)	circRNA	[54]
miR-106b/**93**/25	circ-SMARCA5 (SMARCA5)	circRNA	[55]
**miR-181a-1**/181b-1 **miR-181a-2**/181b-2	CRNDE	antisense RNA	[56]
miR-106a/**18b**/20b/19b-2/92a-2/363	FARSA-AS1	antisense RNA	[57]
**miR-203a**/203b	FBXL19-AS	antisense RNA	[58]
miR-106a/**18b**/20b/19b-2/92a-2/363	FBXW7	lincRNA	[59]
miR-17/**18a**/19a/20a/19b-1/92a-1	FENDRR	lincRNA	[60]
miR-106b/93/**25**	FOXD2-AS1	antisense RNA	[61]
miR-183/96/**182**	GAS5	antisense RNA	[62]
miR-200c/**141** **miR-29b-1**/29a	H19	lincRNA	[63,64]
miR-17/18a/19a/**20a**/19b-1/92a-1	HAND2-AS1	antisense RNA	[65]
miR-106b/**93**/25 **miR-203a**/203b	HOTAIR	antisense RNA	[66,67]
**miR-17**/18a/19a/20a/19b-1/92a-1	HOTAIRM1	lincRNA	[68]
miR-29b-1/**29a**	LIFR-AS1	antisense RNA	[69]
miR-106b/93/**25**	LINC00858	lincRNA	[70]
miR-106b/**93**/25	LINC01567	lincRNA	[71]
miR-203a/**203b**	LINC02595	lincRNA	[72]
**miR-29b-1**/29a	LINC-PINT-variant D	antisense RNA	[29]
**miR-200c/141**	lncRNA ATB	pseudogene	[43,44]
**miR-106b**/93/25 miR-106a/18b/**20b**/19b-2/92a-2/**363**	MALAT1	lincRNA	[73,74,75]
miR-200c/**141**	MEG3	lincRNA	[76]
**miR-203a**/203b	NORAD	lincRNA	[77]
miR-200c/**141**	SNHG15	intronic ncRNA	[78]
miR-17/**18a**/19a/20a/19b-1/92a-1 miR-183/96/**182**	UCA1	lincRNA	[79]
miR-183/96/**182**	XIRP2-AS1	antisense RNA	[80]
miR-106b/**93**/25	XIST	lincRNA	[81]
miR-200c/**141** **miR-181a-1**/181b-1 **miR-181a-2**/181b-2	ZEB1-AS1	antisense RNA	[47,82]

In the first column referring to miRNA clusters, the targeted miRNA is emboldened. Host genes of circular RNAs (circRNAs) are given in parentheses—CircBase [83]. lincRNAs—long intergenic non-coding RNAs.

**Table 3 cancers-13-02979-t003:** Table of miRNA families of selected miRNA clusters and their seed sequences.

Cluster	Family	Seed Sequence
**miR-301b/130b**	miR-130-3p/301-3p/454-3p	AGUGCA
**miR-200c/141**	miR-141-3p/200a-3p	AACACUG
**miR-106a**/18b/**20b**/19b-2/92a-2/363 **miR-106b**/**93**/25 **miR-17**/18a/19a/**20a**/19b-1/92a-1	miR-17-5p/20-5p/93-5p/106-5p/519-3p	AAAGUGC
**miR-181a-1**/**181b-1;181a-2**/**181b-2** **miR-181c**/**181d**	miR-181-5p	ACAUUCA
miR-183/96/**182**	miR-182-5p	UUGGCAA
**miR-183**/96/182	miR-183-5p	AUGGCAC
miR-106a/**18b**/20b/19b-2/92a-2/363 miR-17/**18a**/19a/20a/19b-1/92a-1	miR-18-5p	AAGGUGC
**miR-191**/425	miR-191-5p	AACGGAA
miR-17/18a/**19a**/20a/**19b-1**/92a-1 miR-106a/18b/20b/**19b-2**/92a-2/363	miR-19-3p	GUGCAAA
**miR-203a**/203b	miR-203a-3p.2	UGAAAUG
miR-203a/**203b**	miR-203b-3p	UGAACUG
**miR-222**/**221**	miR-221-3p/222-3p	GCUACAU
miR-452/**224**	miR-224-5p	AAGUCAC
**miR-23a**/27a/24-2	miR-23-3p	UCACAUU
miR-23a/27a/**24-2**	miR-24-3p	GGCUCAG
miR-106b/93/**25** miR-106a/18b/20b/19b-2/**92a-2**/**363** miR-17/18a/19a/20a/19b-1/**92a-1**	miR-25-3p/32-5p/92-3p/363-3p/367-3p	AUUGCAC
miR-23a/**27a**/24-2	miR-27-3p	UCACAGU
**miR-29b-1**/**29a**	miR-29-3p	AGCACCA
miR-191/**425**	miR-425-5p	AUGACAC
**miR-452**/224	miR-452-5p/892-3p	ACUGUUU
miR-183/**96**/182	miR-96-5p/1271-5p	UUGGCAC

In the first column are miRNA clusters with emboldened miRNAs which are included in the miRNA family in the second column. The third column contains the sequence of the seed region common to the miRNA family. Data were extracted from TargetScan [87].

**Table 4 cancers-13-02979-t004:** Changes in miRNA expression in the stools of patients with CRC, compared with controls.

Cluster	miRNA	Levels in Stool
miR-106a/18b/20b/19b-2/92a-2/363	miR-106a	↑ [107,113]
	miR-19b	↑ [114,115]
	miR-92a	↑ [107,114,115,116]
miR-17/18a/19a/20a/19b-1/92a-1	miR-17	↑ [107,115,116]
	miR-18a	↑ [115,117]
	miR-19a	↑ [115]
	miR-20a	↑ [107,114,115]
	miR-19b	↑ [114,115]
	miR-92a	↑ [107,114,115,116]
miR-183/96/182	miR-183	↑ [107]
	miR-96	↑ [107]
miR-222/221	miR-222	↓ [107]
	miR-221	↑ [117]
miR-29b/29a	miR-29b	↓ [117]
	miR-29a	↓ [118]
miR-301b/130b	miR-130b	↑ [119]
miR-452/224	miR-224	↓ [118]

Levels of miRNAs in stool of patients with CRC were compared to samples from healthy donors. ↑ depict increased levels of miRNA in stool of patients with CRC compared to controls, ↓ depicts decreased levels of miRNA.

**Table 5 cancers-13-02979-t005:** miRNAs associated with worse survival or disease recurrence/relapse.

miRNA Cluster	miRNA	Association with Worse Survival	Association with Disease Reccurence/Relapse
miR-106a/18b/20b/19b-2/92a-2/363	miR-106a	↑ TU [104,130,131,132]	↑P [133]; TU [134]
	miR-18b		↑ TU [135]
	miR-92a	↑ S [136], TU [137,138,139] ↓TU [140]	↑ S [141]
	miR-20b	↑ TU [142] ↓ TU [140], P [143]	
	miR-363	↓ S [144]	
miR-106b/93/25	miR-106b	↓ TU [74]	↓ TU [74]
	miR-93	↓ TU [140,145,146,147]	↓ TU [146]
	miR-25	↑ TU [148,149]	
miR-17/18a/19a/20a/19b-1/92a-1	miR-17	↑ TU [150,151,152]; S [104] ↓ TU [140]	↑S [141]
	miR-18a		↑ TU [153]
	miR-19a	↑ S [154]	
	miR-20a	↑ TU [65,130,155]; P [98] ↓ TU [140]	↑ AT [134] ↑T [65]
	miR-19b	↑ LM [156] ↓ TU [140,157]	
	miR-92a	↑ TU [137,138,139]; S [136] ↓ TU [140]	↑ S [141]
miR-181a-1/181b-1 miR-181a-2/181b-2	miR-181a	↑ TU [158,159,160]	
	miR-181b	↑ TU [130]	
miR-181c/181d	miR-181c		↑ TU [135]
	miR-181d	↓ TU [161]	
miR-183/96/182	miR-183	↑ TU [162,163,164]	
	miR-96	↑ TU [165]; P [166] ↓ TU [167]	
	miR-182	↑ TU [165,168,169]	
miR-191/425	miR-191	↓ TU [170]	
miR-200c/141	miR-200c	↑ TU [171,172]; S [173]; P [174] ↓ TU [175,176,177]	
	miR-141	↑ P [166,174,178]; TU [146] ↓ TU [177,179]	↑ TU [146]
miR-203a/203b	miR-203a	↑S [180]; TU [130] ↓ TU [55,181]	↑ AT [134]
miR-222/221	miR-222		↑ TU [135]
	miR-221	↓ TU [182]	↑ TU [135]
miR-23a/27a/24-2	miR-23a	↑ P [98]	
	miR-27a	↑ TU [183,184,185,186]	↑ TU [187]
	miR-24	↑ TU [187] ↓ TU [188]	
miR-29b/29a	miR-29b	↑ TU [189] ↓ P [143];TU [64]	
	miR-29a	↑ TU [190] ↓ TU [191]	↑ P [192] ↓ TU [193]
miR-301b/130b	miR-130b	↑ TU [194]	
miR-452/224	miR-452	↑ TU [195] ↓ TU [196]	
	miR-224	↑ TU [197,198] ↓ TU [182]	↑ TU [197]

↑ indicates increased expression/level of miRNA, ↓ decreased expression/level associated with worse survival or disease recurrence. The source for the measurement of miRNA expression is emboldened—TU (tumour tissue); P (plasma); S (serum); AT (adjacent tissue); LM (liver metastasis).

**Table 6 cancers-13-02979-t006:** List of target genes of up-regulated miRNA clusters.

Cluster	miRNA	Target Genes
miR-106a/18b/20b/19b-2/ 92a-2/363	miR-106a-5p	**ATG7; AMER1; DUSP2;** FOXQ1; **GABBR1**; **TGFBR2**
	miR-106a-3p	**PTEN**
	miR-18b-5p	**CDKN2B**; SOX9
	miR-20b-5p	ADAM9; **GABBR1**; POU5F1
	miR-19b-3p	**SMAD4**
	miR-92a-5p	**AQP8**
	miR-92a-3p	**BCL2L11; DDK3; FBXW7; GSK3β; KLF4; MOAP1; NF2; RBM4; RECK; SMAD7**
	miR-363-3p	EZH2; GATA6; SOX4; SPHK2
miR-106b/25/93	miR-106b-5p	**ATG16L1**; CTSA; **DLC1**; **GABBR1**; ITGB8; **CDKN1A**; **PLK3**; **PRRX1**; **PTEN**; SLAIN2; **TRIM8**
	miR-10b-3p	**DLC1**
	miR-25-5p	NEDD9; PRKCZ
	miR-25-3p	ANGPTL8; **ATXN3**; **MCU**; SEMA4C; SMAD7
	miR-93-5p	ATG12; **ATG16L1**; CCND1; CD274; CTNNB1; **FOXA1**; HIF1A; MYC; **PTEN**; SMAD7
	miR-93-3p	**ARID4B**
miR-17/18a/19a/20a/19b-1/92a-1	miR-17-5p	**BTG3**; **CLU**; CYP7B1; **GABBR1**; **NCOA3**; **PTEN**; **RBL2**; **RND3**; **SIK1** TGFBR2; **TRIM8**; VIM
	miR-17-3p	**F2RL3**
	miR-18a-5p	**ATM**; CCND1; CDC42; CDK19; HIF1A; HNRNPA1; **ING4**; NEDD9; **PIAS3**; **SMAD4**; TBPL1
	miR-19a-3p	**FOXF2**; KRAS; **PTEN**; TF; TGFBR2; **TGM2**; **THBS1**; **TIA1**; **TNFAIP3**
	miR-20a-5p	**AMER1**; **ATG5**; **BID**; **BNIP2**; **CDKN1A**; CXCL8; FOXJ2; **GABBR1**; **MAP3K5**; **MICA**; **PDCD4**; **RB1CC1**; **SMAD4**; TGFBR2; VEGFA
	miR-19b-3p	**SMAD4**
	miR-92a-5p	**AQP8**
	miR-92a-3p	**BCL2L11**; **DDK3**; **FBXW7**; **GSK3β**; **KLF4**; **MOAP1**; **NF2**; **RBM4**; **RECK**; **SMAD7**; **PTEN**
miR-181a-1/181b-1miR-181a-2/181b-2	miR-181a-5p	CTNNB1; E2F5; MMP14; PLAG1; **PTEN**; **SRCIN1**; **STAT1**; TCF4; **WIF1**
	miR-181b-5p	**PDCD4**; **PIAS3**; RASSF1; TUSC3
miR-181c/181d	miR-181d-5p	PEAK1; **FBXL3**; **CRY2**
miR-183/96/182	miR-183-5p	**ABCA1**; **AFDN**; **AKAP12**; **ATG5**; **RCN2**; **UVRAG**
	miR-96-5p	CCND1; **FOXO1**; **FOXO3a**; GPC1; KRAS; **TP53INP1**; **TPM1**
	miR-96-3p	**RECK**
	miR-182-5p	**AGER**; **DAB2IP**; **FBXW7**; **FOXF2**; **FOXO1**; **FOXO3a**; MTDH; MYC; **SATB2**; **ST6GALNAC2**; **THBS1**
miR-191/425	miR-191-5p	**C/EBPβ**; **TIMP3**
	miR-425-5p	**CTNND1**; **PDCD10**
miR-200c/141	miR-200c-3p	BMI1; CDK2; ETS1; FLT1; KLF14; SOX2; **VLDLR**; ZEB1
	miR-200c-5p	PRICKLE2
	miR-141-3p	CCND2; **DLC1**; EGFR; GEMIN2; **MAP2K4**; MAP4K4; PDCD4; SIRT1; TRAF5; ZEB1
miR-203a/203b	miR-203a-3p	AKT2; **ATM**; CPEB4; CREB1; CSE1L; CTNNB1; DNMT3b; EIF5A2; GATA6; NEDD9; **PDE4D**; RNF6; ROBO1; SIK2; TCF4; THBS2; TLE5; TYMS; KLK6
	miR-203b-3p	BCL2L1
miR-221/222	miR-221-5p	MBD2
miR-23a/27a/24-2	miR-23a-3p	**ABCF1**; **APAF1**; **MARK1**; **PDK4**; **SEMA6D**
	miR-27a-3p	**BTG1**; **CALR**; **FAM172A**; **RXRα**; **SFRP1**; SGPP1; SMAD2; **TP53**; VANGL
	miR-24-5p	CTNNB1
	miR-24-3p	DHFR; DND1; VHL
miR-29b-1/29a	miR-29b-3p	BCL9L; GRN; SIRT1; TIAM1
	miR-29b-5p	**SMAD3**
	miR-29a-3p	**KLF4**; MMP2; **PTEN**; RPS15A; TNFAIP3
miR-301b/130b	miR-130b-3p	ITGB1; ITGA5
miR-452/224	miR-224-5p	**CDH1**; **CDKN1A**; CDS2; **GSK3β**; LGALSL; MBD2; **PHLPP1**; **PHLPP2**; **SFRP2**; SLC4A4; **SMAD4**; **USP3**
	miR-452-3p	**GSK3β**

Genes with tumour suppresor function are bolded in this table.

**Table 7 cancers-13-02979-t007:** Panther pathways significantly increased in the statistical overrepresentation test according to target gene miRNAs of miRNA clusters.

Pathway	Pathway Code	miRNA Cluster
Angiogenesis	P00005	miR-106b/93/25 miR-17/92a-1
Apoptosis signaling pathway	P00006	miR-203a/203b
CCKR signaling map	P06959	miR-106b/93/25 miR-17/92a-1 miR-181a-1/181b-1; miR-181a-2/181b-2 miR-203a/203b
Gonadotropin-releasing hormone receptor pathway	P06664	miR-17/92a-1 miR-200c/141
Hypoxia response via HIF activation	P00030	miR-106b/93/25 miR-17/92a-1
Integrin signalling pathway	P00034	miR-17/92a-1 miR-301b/130b
Interferon-gamma signaling pathway	P00035	miR-181a-1/181b-1; miR-181a-2/181b-2
Interleukin signaling pathway	P00036	miR-17/92a-1
JAK/STAT signaling pathway	P00038	miR-181a-1/181b-1; miR-181a-2/181b-2
p53 pathway	P00059	miR-17/92a-1 miR-203a/203b
p53 pathway feedback loops 2	P04398	miR-106b/93/25 miR-17/92a-1 miR-181a-1/181b-1; miR-181a-2/181b-2 miR-203a/203b
PI3 kinase pathway	P00048	miR-17/92a-1 miR-183/96/182
Ras Pathway	P04393	miR-17/92a-1
TGF-beta signaling pathway	P00052	miR-106a/363 miR-17/92a-1
VEGF signaling pathway	P00056	miR-17/92a-1
Wnt signaling pathway	P00057	miR-106b/93/25

## Data Availability

Data is contained within the article or Appendix A.

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
