# Peer review of "miRNA Clusters with Up-Regulated Expression in Colorectal Cancer"

_cancers, 2021, doi:10.3390/cancers13122979_

Round 1

Reviewer 1 Report

Review by Pidikova and Herichova represents a valuable work summarizing intensive search and attempts of several research groups worldwide to identify key specific biomarkers reliable for early non-invasive diagnostics of CRC, still one of the top most common cancers in Europe and North America.

By extensive analysis of published experimentally validated data, authors identify a reasonable number of up-regulated non-coding RNAs – miRNAs that have capacity to serve as potential biomarkers of CRC. 15 miRNA clusters and 181 protein-coding genes targeted by members of these clusters were identified. Using GO-Slim and Panther analyses, target genes were classified according to molecular function, biological processes, protein classes and pathways. Several highly relevant overrepresented categories have been revealed, pointing on strong association of selected miRNAs with occurrence of metastasis and poor patient survival.

Works of this type usually face the problem of summarizing results of miRNA expression analysis and evaluation obtained with different methodological approaches in individual studies. While used methods of miRNA expression analysis were shown in Table S1, similarly, used reference genes and involved healthy controls descriptions could have been summarized and provided.

The main objection to this review concerns many inconsistencies in citations of references. For example, as randomly checked:

Lane 326 – [94] cited in relation to miR-182

  1. Li, L.; Guo, Y.; Chen, Y.; Wang, J.; Zhen, L.; Guo, X.; Liu, J.; Jing, C. The DiagnosticEfficacy and Biological Effects of MicroRNA-29b for Colon Cancer. Technology in Cancer Research and Treatment 2016, 15, 772–779, doi:10.1177/1533034615604797.

Lane 428 – Table 4:

[132] cited in relation to miR-106a

  1. Diaz, T.; Tejero, R.; Moreno, I.; Ferrer, G.; Cordeiro, A.; Artells, R.; Navarro, A.;Hernandez, R.; Tapia, G.; Monzo, M. Role of MiR-200 Family Members in Survival of Colorectal Cancer Patients Treated with Fluoropyrimidines. Journal of Surgical Oncology 2014, 109, 676–683, doi:10.1002/jso.23572.

 [148] cited in relation to miR-363

  1. Xiao, Z.-G.; Deng, Z.-S.; Zhang, Y.-D.; Zhang, Y.; Huang, Z.-C. Clinical Significanceof MicroRNA-93 Downregulation in Human Colon Cancer. European Journal of Gastroenterology and Hepatology 2013, 25, 296–301, doi:10.1097/MEG.0b013e32835c077a.

[151] cited in relation to miR-93

  1. Li, X.; Yang, C.; Wang, X.; Zhang, J.; Zhang, R.; Liu, R. The Expression of MiR-25 Is Increased in Colorectal Cancer and Is Associated with Patient Prognosis. Medical Oncology 2014, 31, doi:10.1007/s12032-013-0781-7.

[193] cited in relation to miR-24

  1. Inoue, A.; Yamamoto, H.; Uemura, M.; Nishimura, J.; Hata, T.; Takemasa, I.; Ikenaga, M.; Ikeda, M.; Murata, K.; Mizushima, T.; et al. MicroRNA-29b Is a Novel Prognostic Marker in Colorectal Cancer. Annals of Surgical Oncology 2015, 22, 1410–1418,doi:10.1245/s10434-014-4255-8.

[195] cited in relation to miR-29b

  1. Gao, Y.; Liu, Y.; Du, L.; Li, J.; Qu, A.; Zhang, X.; Wang, L.; Wang, C. Down-Regulation of MiR-24-3p in Colorectal Cancer Is Associated with Malignant Behavior. Medical Oncology 2015, 32, doi:10.1007/s12032-014-0362-4.

[147] cited in relation to miR-29b

  1. Zhuang, M.; Zhao, S.; Jiang, Z.; Wang, S.; Sun, P.; Quan, J.; Yan, D.; Wang, X. MALAT1 Sponges MiR-106b-5p to Promote the Invasion and Metastasis of Colorectal Cancer via SLAIN2 Enhanced Microtubules Mobility. EBioMedicine 2019, 41, 286–298, doi:10.1016/j.ebiom.2018.12.049.

[199] cited in relation to miR-29a

  1. Colangelo, T.; Fucci, A.; Votino, C.; Sabatino, L.; Pancione, M.; Laudanna, C.; Binaschi, M.; Bigioni, M.; Alberto Maggi, C.; Parente, D.; et al. MicroRNA-130B Promotes Tumor Development and Is Associated with Poor Prognosis in Colorectal Cancer. Neoplasia (United States) 2013, 15, 1218–1231, doi:10.1593/neo.13998.

[200] cited in relation to miR-29a

  1. Li, T.; Jian, X.; He, H.; Lai, Q.; Li, X.; Deng, D.; Liu, T.; Zhu, J.; Jiao, H.; Ye, Y.; et al.MiR-452 Promotes an Aggressive Colorectal Cancer Phenotype by Regulating a Wnt/β-Catenin Positive Feedback Loop. Journal of Experimental and Clinical Cancer Research 2018, 37, doi:10.1186/s13046-018-0879-z.

[201] cited in relation to miR-130b

  1. Yan, J.; Wei, R.; Li, H.; Dou, Y.; Wang, J. MiR-452-5p and MiR-215-5p Expression Levels in Colorectal Cancer Tissues and Their Relationship with Clinicopathological Features. Oncology Letters 2020, 20, 2955–2961, doi:10.3892/ol.2020.11845.

 Lane 561 -[231] cited in relation to miR-96-5p and miR-182-5p

  1. Jiang, T.; Ye, L.; Han, Z.; Liu, Y.; Yang, Y.; Peng, Z.; Fan, J. MiR-19b-3p Promotes Colon Cancer Proliferation and Oxaliplatin-Based Chemoresistance by Targeting SMAD4: Validation by Bioinformatics and Experimental Analyses. Journal of Experimental and Clinical Cancer Research 2017, 36, doi:10.1186/s13046-017-0602-5.

Lane 566 -[235] cited in relation to miR-93-5p

  1. Wang, Y.; Ren, J.; Gao, Y.; Ma, J.Z.I.; Toh, H.C.; Chow, P.; Chung, A.Y.F.; Ooi, L.L.P.J.; Lee, C.G.L. MicroRNA-224 Targets SMAD Family Member 4 to Promote Cell Proliferation and Negatively Influence Patient Survival. PLoS ONE 2013, 8, doi:10.1371/journal.pone.0068744.

Lane 578-[240] cited in relation to miR-96-5p

  1. Zhou, Y.; Wan, G.; Spizzo, R.; Ivan, C.; Mathur, R.; Hu, X.; Ye, X.; Lu, J.; Fan, F.; Xia, L.; et al. MiR-203 Induces Oxaliplatin Resistance in Colorectal Cancer Cells by Negatively Regulating ATM Kinase. Molecular Oncology 2014, 8, 83–92, doi:10.1016/j.molonc.2013.09.004.

All citations within the tables and text need to be checked.

Minor comments:

Lane 166 - typo: “clinicopathological” or “clinico-pathological” – should be used uniformly within the text

Lane 187 – typo: “miRNA expression is thus frequently correlated”

Lane 319 – “A list of relative expression levels of the analysed miRNA clusters in the circulation is in Table S3.”…relative expression levels are not shown in this table, only increase or decrease of expression

Lane 453 - …”inhibit DNA replication and translation through the formation of DNA adducts.” – Probably, DNA replication and transcription was meant…

Lane 508 - Figure 2 – “The significantly overrepresented term is underlined (P˂0.001; FDR ˂0.001)” – but no term is underlined in this figure

Reviewer 2 Report

The manuscript submitted by Pidíková and Herichová is a systematic review of studies on microRNAs (miRNAs) upregulated in colorectal cancer (CRC), which could be addressed in the context of miRNA clinical use as prognostic and diagnostic non-invasive biomarkers. The subject of this review is interesting, but it should be presented more clearly and in more details with the focus of the impact of miRNA use in the clinical setting. Additionally, several systematic reviews have been published in the last couple of years on the role of miRNAs in colorectal cancer, thereby weakening the novelty of this manuscript.

Other major comments are:

  1. Consider reducing the number of references as this would improve the read flow and strengthen the focus of the review. Additionally, less than 20% of the refences are from publications in the last 2 years. Recent references would show that this subject is of current interest and could bring relevant contribute to molecular biology and cancer research.
  2. Confusion with ncRNAs classification; small ncRNAs are not all miRNAs!
  3. Confusion on some of the exclusion criteria: “2. studies analysing data from public databases and datasets (like the GEO database)” and “4. studies where only plasma or serum samples from patients with CRC were analysed”. Why are they exclusion criteria? Please explain this and consider that including these studies might enforce the study. Public datasets have a massive content of information which seem essential in the context of a systematic review to analyse miRNAs in CRC as diagnostic and prognostic biomarkers or therapeutic targets. Additionally, miRNAs are well studied for their use as diagnostic biomarkers in cancer patient biological fluids (e.g., plasma and serum) so the criteria 4 might exclude a huge number or fundamental data in this context.
  4. LncRNA-miRNA correlation and interaction should be discussed more in depth, and the authors should consider including research on lncRNAs as precursors of mature miRNAs rather than only as miRNA-sponges.
  5. More clarity and background discussion are needed on the current progress of the use of miRNAs (and more broadly ncRNAs) in the clinics.
  6. For the sentences: “The transcription factor MYC regulates clusters miR-29b-1/29a [258], miR-17/92a-1 [50,51,259,260]and miR-181d [52]. Increased expression of MYC in tumour tissue was associated with improved survival of patients with CRC [261] however, a meta-analysis from 2018 revealed that MYC expression was not associated with patient survival [262] therefore, more research is needed to identify conclusively the role of this transcriptional factor in miRNA632 mediated progression of CRC”; MYC is often studied as an oncogene in CRC, therefore the increased expression in tumour tissue, which is associated with better prognosis, should be discussed more in this context. Are there other analyses showing MYC involvement in CRC survival? Since MYC is often not an optimal therapeutic target for cancer treatment, investigating downstream MYC-regulated miRNA clusters, might be revealed as successful therapeutic strategy if further investigated. The authors should discuss more on these aspects and possibly present and analyse more data in this context.
  7. Since lncRNAs and epigenetics are both discussed as miRNA interactors, it would add novelty to this review to deepen into knowledge on a possible “triangular” interaction in which the triple detection of specific lncRNAs and epigenetic changes together with miRNAs, previously studied as mutual interactors, could add strength to potential non-invasive diagnostic and prognostic approaches in CRC.
  8. Acronyms should be better introduced for clarity.
  9. Consider syntactic language revision.

Reviewer 3 Report

In their interesting review paper the authors summarize the knowledge about miRNAs in colon cancer using their own analysis pipeline for searching articles about this topic. They give their own analyses about their associations with biological prcesses or targeted protein calsses etc. 

I feel the heading "miRNA clusters ... and their target genes" promises at least 50% of the article should give an overview about genes and their functions, but the information is only given in the text and in supplementary figures. The authors could change the title, then its fine or they could give a new table with most interesting gene targets summarized according to their functions. 

Round 2

Reviewer 2 Report

The revision version of the manuscript submitted by Pidíková and Herichova is improved but there are still some last concerns to be considered, before publication.

  1. Further reduction in references number (especially for the less recent ones) would be beneficial for the read-flow. Less than 400 references are considered according to the criteria defined by the authors, but the total number of references is still almost 550.
  2. The sentence: “The present systematic review is focused on clusters of micro RNAs (miRNAs) with 51 up-regulated expression in colorectal cancer (CRC) tissue, compared with adjacent tissue. 52 miRNAs are class of small non-coding RNAs with length approximately 20 bp that show 53 the potential to be used as diagnostic tools” is repeated twice in the introduction.
  3. More clarity is needed throughout the manuscript on regulation between molecules. Positive (e.g. upregulation) or negative (e.g. downregulation) regulations should be specified, rather than simple “regulation”.
  4. Consider further syntactic language revision.

Author Response

Reviewer 2

Comments and Suggestions for Authors

Dear reviewer,

thank you very much for your valuable time and your comments, suggestions and useful advice. We implemented your suggestion as much as it was possible to keep integrity of systematic review using defined methodology. We hope, you will find revised version of MS acceptable.

Further reduction in references number (especially for the less recent ones) would be beneficial for the read-flow. Less than 400 references are considered according to the criteria defined by the authors, but the total number of references is still almost 550.

We deeply apologise for misunderstanding about number of references identified by search. Above mentioned calculation of references to be used in MS relay on our scheme in figure 1. According figure 1 screening identified 399 articles from which 47 of screenings were accepted as a basis for the systematic review. This process is described in methods in the chapter 2.1.

However, in chapter 2.2 of Methods we describe the second round of search that was not shown in figure 1 (described in the chapter “methods”) that resulted in the implementation of 418 articles providing information about miRNA levels in human CRC tissues and the circulation and/or stool. This bunch of papers is a source of information about associations of miRNA expression with clinicopathological characteristics and validated miRNA target genes. These articles are also part of systematic search and issue form the methodology that has already been approved by reviewers 1 and 3.

To elucidate and explain methodology we included information about number of articles accepted in the second round of screening into chapter 2.2. Similarly, we changed figure 1 to show whole methodology in the graph. Changed figure is also included in this letter.

In spite of the above mentioned arguments we tried our best to find some articles that could be omitted without destroying MS integrity and excluded 57 articles that were usually older than 3 years. Time distribution of omitted and incorporated articles is shown in graph included in this letter. Only 26 articles are used in the Introduction and introductory section of chapters 3.1 – 3.8.

Unfortunately, we cannot exclude 150 articles, as requested, as in that case this MS would not be systematic review anymore. It would be just review of randomly selected articles and information provided for reader would not be supported by complete search as it is indicated by the introduction.

We deeply apologise for misunderstanding with figure 1 and chapter 2.2 in methods again.

Graph showing time distribution of articles used in MS.

Changed Figure 1. Final number of articles selected by search is 47 + 418.

The sentence: “The present systematic review is focused on clusters of micro RNAs (miRNAs) with 51 up-regulated expression in colorectal cancer (CRC) tissue, compared with adjacent tissue. 52 miRNAs are class of small non-coding RNAs with length approximately 20 bp that show 53 the potential to be used as diagnostic tools” is repeated twice in the introduction.

Thank you for the comment, text was corrected.

More clarity is needed throughout the manuscript on regulation between molecules. Positive (e.g. upregulation) or negative (e.g. downregulation) regulations should be specified, rather than simple “regulation”.

We are very grateful for this comment.

Term “regulation” has been replaced in:

- chapter 3.2 – the last sentence

- chapter 3.3 – the 3rd and 4th paragraphs

- chapter 3.4. – the first paragraph

- chapter 3.8 – the 3rd paragraph, 10th paragraph

- discussion – the 3rd paragraph, 4th paragraph, 6th paragraph and 13th paragraph.

All changes are indicated by yellow colour in the text.

Consider further syntactic language revision.

Thank you for the comment, MS was carefully checked again.